# Quantitative Measurement of Spinal Cerebrospinal Fluid by Cascade Artificial Intelligence Models in Patients with Spontaneous Intracranial Hypotension

**DOI:** 10.3390/biomedicines10082049

**Published:** 2022-08-22

**Authors:** Jachih Fu, Jyh-Wen Chai, Po-Lin Chen, Yu-Wen Ding, Hung-Chieh Chen

**Affiliations:** 1Computer Aided Measurement and Diagnostic Systems Laboratory, Department of Industrial Engineering and Management, National Yunlin University of Science and Technology, Yunlin 640, Taiwan; 2Department of Radiology, Taichung Veterans General Hospital, Taichung 407, Taiwan; 3Department of Radiology, College of Medicine, National Chung Hsing University, Taichung 402, Taiwan; 4Department of Radiology, College of Medicine, China Medical University, Taichung 404, Taiwan; 5Department of Neurology, Taichung Veterans General Hospital, Taichung 407, Taiwan; 6Department of Radiology, School of Medicine, National Yang-Ming Chiao-Tung University, Taipei 112, Taiwan

**Keywords:** SIH, CSF segmentation, deep learning, cascade AI model, object detection, semantic segmentation

## Abstract

Cerebrospinal fluid (CSF) hypovolemia is the core of spontaneous intracranial hypotension (SIH). More than 1000 magnetic resonance myelography (MRM) images are required to evaluate each subject. An effective spinal CSF quantification method is needed. In this study, we proposed a cascade artificial intelligence (AI) model to automatically segment spinal CSF. From January 2014 to December 2019, patients with SIH and 12 healthy volunteers (HVs) were recruited. We evaluated the performance of AI models which combined object detection (YOLO v3) and semantic segmentation (U-net or U-net++). The network of performance was evaluated using intersection over union (IoU). The best AI model was used to quantify spinal CSF in patients. We obtained 25,603 slices of MRM images from 13 patients and 12 HVs. We divided the images into training, validation, and test datasets with a ratio of 4:1:5. The IoU of Cascade YOLO v3 plus U-net++ (0.9374) was the highest. Applying YOLO v3 plus U-net++ to another 13 SIH patients showed a significant decrease in the volume of spinal CSF measured (59.32 ± 10.94 mL) at disease onset compared to during their recovery stage (70.61 ± 15.31 mL). The cascade AI model provided a satisfactory performance with regard to the fully automatic segmentation of spinal CSF from MRM images. The spinal CSF volume obtained through its measurements could reflect a patient’s clinical status.

## 1. Introduction

Spontaneous intracranial hypotension (SIH) is caused by cerebrospinal fluid (CSF) leakage through a spinal dura defect, leading to CSF hypovolemia. The incidence of SIH is 5 per 100,000/year, with a female-to-male ratio of 2:1. Diagnosis of SIH depends on typical clinical presentation, such as orthostatic headache and typical radiological findings [1,2]. Orthostatic headache occurs when hypovolemic CSF cannot provide sufficient buoyancy; then, the pain-sensitive structure is stretched by descended brain tissue. Typical radiological abnormalities include the direct visualization of CSF leaks using magnetic resonance myelography (MRM) [3,4] or computed tomographic myelography (CTM) [5] and diffused pachymeningeal enhancement (DPE), pituitary hyperemia, dural sinus engorgement, subdural effusion (SDE), subdural hematoma (SDH), or brain descent in conventional brain magnetic resonance imaging (MRI) and epidural venous plexus engorgement in spinal MRI [6,7].

Epidural blood patch (EBP) is the treatment of choice for SIH patients who have been nonresponsive to conservative hydration treatment. Evaluation of disease severity and treatment response usually relies on the grading of headache scores, the visual analog scale (VAS), and MRI findings [8]. Researchers have used imaging signs to evaluate the severity of CSF hypovolemia—for example, whether the number of neural sleeves leaking CSF has grown, i.e., spinal CSF leakage numbers, or whether there was the presence of qualitative radiological abnormalities such as DPE or SDH [9]. However, headache scores are usually subjective and can vary between individuals. Qualitative radiological findings and semi-quantitative measurement of spinal CSF leakage numbers might not accurately reflect subtle disease changes. Therefore, quantifying spinal CSF volume is crucial to evaluating the hypovolemic status of SIH patients. The spinal MRM images used for assessing the CSF volume usually contain over 1000 isotropic images for each subject. Our previous study established quantitative CSF measurement using an entropy-based segmentation method with good disease status correlation [10]. However, the segmentation performance of the entropy-based method was not objectively evaluated. In recent years, due to the rapid advancement of artificial intelligence (AI) for medical imaging analysis, we are keen to understand the effectiveness of using AI to measure spinal CSF volume in patients with SIH quantitatively.

Krizhevsky et al. developed deep convolutional neural network architectures and formed a team (SuperVision) that won the ImageNet competition in 2012 by a large margin [11]. Since then, deep learning has drawn great attention and grown exponentially. Ronneberger et al. applied U-net to medical image segmentation and won the ISBI cell-tracking challenge [12]. U-net and its variants provide many medical image applications, such as the segmentation of brain tumor and tissue, stroke lesions, the cardiovascular structure, spinal cord, and various types of cancer [13]. For spine-related MR image segmentation, there are a couple of published studies that use U-net to segment the spinal cord [14,15]. However, to date, there is no literature in the field of deep learning-based spinal CSF segmentation.

In addition to semantic segmentation, border detection is the crucial function for fully automated computer-aided diagnosis. Redmon et al. created You Only Look Once (YOLO) in 2016, and it has been widely used for medical image studies [16].

In this study, we proposed a cascade architecture that could individually perform the object detection task and the semantic segmentation task. Then, two types of aforementioned output were fused by a union operation. We expected the proposed cascade AI to provide both object detection and semantic segmentation functions. Meanwhile, it may provide satisfactory segmentation performance to measure spinal CSF volume and evaluate CSF hypovolemia in SIH patients. Furthermore, we investigated whether the segmented spinal CSF volume would be consistent with other MR findings and correlate with disease status.

## 2. Materials and Methods

### 2.1. Enrolled Subjects

From January 2014 to December 2019, patients who had been diagnosed with spinal CSF leakage and received treatment to complete recovery at our hospital were retrospectively reviewed. They were assessed using MRI and MRM at disease onset and at their complete recovery stage, which was defined as there being neither clinical symptoms nor abnormal radiological findings. Patients with poor-quality MR images were excluded. The definition of poor-quality images was that the scan area did not include the whole spinal canal, uneven signal intensity through the spinal canal, or presence of large imaging artifacts. Healthy volunteers (HVs) with no known spinal disease were recruited and received MRM examination.

Twenty-five subjects (13 SIH patients and 12 age- and gender-matched HVs, Appendix A) were enrolled to train and test the performance of object detection and image segmentation. Another 13 patients were enrolled in the cohort study to measure the spinal CSF volume at disease onset and at complete recovery to determine whether the measured CSF volume could accurately reflect the clinical condition in the real world. Table 1 shows the demographic data of the subjects enrolled in this study.

This study was approved by the local institutional review board (SE17334A, CE16103A, CE22242B), and informed consent was obtained.

We divided the image data of 25,603 slices of MRM images acquired from 25 subjects (13 patients and 12 HVs) into training and validation datasets and a test dataset with a ratio of 4:1:5. The training dataset contained 7192 slices and 6201 slices of annotated MR images from 7 patients and 6 HVs, respectively. Overall, 13,393 slices of annotated MR images were used to train and validate the proposed model. The test dataset contained 6332 slices and 5878 slices of annotated MR images from 6 patients and 6 HVs, respectively. Overall, 12,210 slices of annotated MR images were used to test the model performance. The ratio of images used in the training phase and test phase was 52%:48% (13,393:12,210).

### 2.2. Neuroimaging Acquisition

A 1.5 T MRI scanner (MAGNETOM Aera, Siemens Healthcare, Erlangen, Germany) was used. Whole-spine MRM was performed in all patients at disease onset and at complete recovery and in HVs using three-dimensional sampling perfection and optimized contrast using different flip-angle evolution (3D-SPACE) sequences. The MRI parameters were as follows: TR = 3000 ms, TE = 560 ms, isotropic voxel size = 0.9 mm^3^, matrix size = 320 × 320 pixels, and field of view (FOV) = 200 mm. Fat suppression and generalized autocalibrating partially parallel acquisition (GRAPPA) imaging reconstruction with an acceleration factor of two were used. Images were obtained volumetrically in the coronal plane parallel to the cervical-to-thoracic and thoracic-to-lumbar regions of the spine, and then axial multi-planar reformation (MPR) images were reconstructed with a slice thickness of 5 mm for spinal CSF leakage number assessment.

### 2.3. Object Detection and Semantic Segmentation of Spinal CSF

Figure 1 shows the flow of the cascade AI model. Two modules independently analyzed the same input, and final CSF output was made by combining the output of two modules with the objective of reducing the model error. The module of object detection and the module of semantic segmentation were combined using Boolean AND (**∩**) operation.

This paper divides the tasks into objection detection and semantic segmentation. Therefore, the conventional convolution neural networks (CNNs), designed for classification problems [17], are not used in this paper. For the object detection module, YOLO v3 was prevalent, based on a web search performance [16]. YOLO v3 was chosen in our object detection module (Module 1) to detect the location of hyperintense CSF and then generate the area of ROI. For semantic segmentation, U-net and variants had been applied in many medical image segmentations [13]. In traditional U-net, the feature maps of the contracting path are directly concatenated onto the corresponding layers in the expansive path. U-net++ uses skip connections as an intermediary grid between the contracting and expansive paths [18]. The skip connection mechanism aids U-net++ by propagating more semantic information between the two paths, enabling it to segment images more accurately than U-net [13]. In this study, both U-net and U-net++ were used in the semantic module (Module 2) to segment CSF from the background tissue. The architectures of YOLO v3, U-net, and U-net++ are shown in Figure 2.

YOLO v3 applied multi-scale predictions and a better backbone classifier to increase performance by increasing the number of bounding boxes and to enhance the ability to detect small objects. The details of YOLO v3 can be found in [19,20]. The U-net framework can be viewed as an encoder-decoder architecture. A U-net consists of a contracting path (left-hand side of the U-net) and an expansive path (right-hand side of the U-net). The spatial information is reduced, while feature information is increased, in the contracting path. The expansive pathway includes a sequence of up-convolutions and skip connections (also called shortcut connections) to combine the feature and spatial information [12]. U-net++ is the nested architecture of U-net. Compared with U-net, U-net++ generates full-resolution feature maps at multiple semantic levels. The details of U-net++ can be found in [18]. The sets of hyperparameters applied to the networks in this paper are as follows. For YOLO v3, the input image size is 128 × 128. Batch size and optimizer are set to be 32 and stochastic gradient descent, respectively. Loss functions combine mean square error (MSE) and binary cross entropy. For U-net and U-net++, batch size and optimizer are set to be 7 and Adam, respectively. The loss function is binary cross entropy.

### 2.4. Performance Evaluation

The gold standard of spinal CSF from MRM images, separately segmented by two experts, was first manually segmented by a medical researcher and further verified by an experienced neuroradiologist. The segmentation performance of the cascade YOLO and U-net++ AI model was measured by the metric of intersection over union (IoU), also named Jaccard similarity, which measures the ratio of the intersection of the samples to their union. The IoUs of four combination modules, cascade of U-net, cascade U-net++, U-net++, and U-net, were calculated and compared.

### 2.5. Spinal CSF Quantification in Cohort Study

The AI model with the best performance was used to automatically quantify the spinal CSF volume of the other 13 SIH patients using their MRI at disease onset and after complete recovery. The spinal CSF leakage numbers for each patient at disease onset were recorded. Most of the patients received intravenous hydration for 3 days and target EBP near the spinal leakage site as the treatment method.

### 2.6. Statistical Analysis

Statistical analyses were performed in SPSS statistical analysis software, version 22.0 (IBM SPSS statistics, Chicago, IL, USA). The IoU between different AI models and the difference between spinal CSF before and after treatment were compared using the paired *t*-test. The correlation of the spinal CSF leakage numbers and the measured CSF volume at disease onset was analyzed using Pearson correlation. Further, *p*-values less than 0.05 were indicative of statistical significance.

## 3. Results

### 3.1. Object Detection and Semantic Segmentation of Spinal CSF

Since the training dataset focused on the spine, detection performance decays when the tested images are located above the craniocervical junction. In other images in the spine region, the performance of detection and the segmentation of spinal CSF was quite satisfactory. Figure 3a,b show a successful and a failed example caused by poor objection detection. The upper border of this study’s region of interest (ROI) is the craniocervical junction, and those failed cases do not influence the performance.

Table 2 shows the segmentation performance in terms of IoU from the test dataset, which contains 12 subjects (6 patients and 6 HVs) with 12,210 slices of MR images.

The results show that the performance from cascade of U-net++ and YOLO v3, from cascade of U-net and YOLO v3, from U-net++, and from U-net on average were 0.9374, 0.9373, 0.9102, and 0.9077, respectively. Single-tailed paired *t*-tests were conducted to determine whether the cascade models significantly outperformed the non-cascade models in terms of IoU. The test was also conducted to determine whether U-net++ significantly outperformed U-net in the non-cascade modes. Therefore, we tested the hypothesis:

Equation (1): H_0_: IoU_1_ = IoU_2_

Equation (2): H_1_: IoU_1_ > IoU_2_

For the comparison of algorithms in non-cascade models, IoU_1_ is the IoU between U-net++ and the gold standard; IoU_2_ is the IoU between U-net and the gold standard. For the cascade models vs. non-cascade models, IoU_1_ is the IoU between U-net++ (or U-net) integrated with YOLO v3 and the gold standard; IoU_2_ is the IoU between Unit++ (or U-net) and the gold standard. Table 3 shows the output of the paired *t*-test at a significance of α = 5%. Since 12,210 slices of MR images were enrolled in each type of model, the degree of freedom (DoF) was 12,209 in the paired *t*-test. A confidence interval with a lower limit greater than 0 implies that the null hypothesis H_0_ should be rejected. In non-cascade models, acceptance of the alternative hypothesis (H_1_) implies that U-net++ performs significantly better in segmentation than U-net. For the cascade models vs. non-cascade models, acceptance of the alternative hypothesis (H_1_) implies that the cascade models significantly outperform the corresponding non-cascade models (U-net++ or U-net). The experimental results showed that the proposed cascade models, the integration of object detection and semantic segmentation, provided more refined segmentation output than only semantic segmentation modules (non-cascade models).

### 3.2. Cohort Study

We applied the YOLO v3 plus Unet++ to the MRM images at the disease onset and complete recovery stages in 13 SIH patients. Significantly decreased spinal CSF volume during disease onset compared to recovery stage (59.32 ±10.94 mL vs. 70.61 ± 15.31 mL, *p* < 0.001) was found. The spinal CSF volume during disease onset was negatively correlated with spinal leakage numbers (correlation coefficient −0.628, *p* =0.016). The mean spinal CSF leakage number was 13.79 ± 7.69 at disease onset.

## 4. Discussion

Hypovolemia is the key pathophysiology in SIH. Semi-quantitative measurement or quantitative measurement of spinal CSF amount was proved to be an effective method in evaluating treatment response in previous studies [9,10]. By simply counting the numbers of spinal CSF leakages, a higher spinal CSF leakage number was found in post-EBP MRI, and the possibility of treatment failure increased [9], i.e., the numbers of spinal CSF leakages could reflect the disease severity. On the other hand, in our previous study, we found that CSF obtained using entropy-based image segmentation could acceptably reflect the disease status of SIH patients [10]. Since no gold standard was applied in that research, the accuracy of the entropy-based image segmentation method was not objectively evaluated. Our preliminary study showed that the deep learning AI method outperformed the aforementioned conventional entropy-based image segmentation method [21].

In the current study, we used the trained cascade semantic segmentation model to obtain spinal CSF volume in SIH patients. The results showed that cascade models significantly outperformed the non-cascade models. Among the four combinations of modules, the IoU of the cascade of YOLO v3 and U-net++ (0.9347) was the highest. The proposed cascade of YOLO v3 and U-net++ model took less than 45 s on average for 1130 slices of MR images (13,559 slices divided by 12 persons in the test dataset) of each patient by using GUP Nvidia-GeForceRTX2080 Ti with 32 GB of RAM.

U-net and the variants are widely used in medical image data, for example, segmentation of brain tumor and tissue, stroke lesion, cardiovascular structure, spinal cord, and various types of cancer in recent years [13]. In our study, IoU values of U-net and U-net++ were good for spinal CSF segmentation, but the performance was better for combing object detection and semantic segmentation. Jaeger et al. developed Retina U-Net, fusing the Retina Net one-stage detector with the U-net architecture to detect and segment lung lesions in CT images and breast lesions in diffusion MR images [22]. Compared with U-net providing the function of semantic segmentation only, Retina U-Net provides both object detection and semantic segmentation. Our preliminary study applied both U-net and Retina U-Net to quantify spinal CSF in MRM images, and the experimental results showed that the Retina U-Net provided a lower segmentation performance than U-net [23]. This means that Retina U-Net offers the object detection function by sacrificing semantic segmentation performance.

Therefore, we used YOLO v3 as our object detection module. After Redmon et al. first introduced the YOLO algorithm, it drew wide attention because of its speed and accuracy [16]. Since then, YOLO has been widely used in various applications and subsequently evolved advanced versions. YOLO v2 improved the inaccurate positions and low recall rate of YOLO. Compared with YOLO v2, YOLO v3 provides multi-scale features. YOLO v4 was developed for small object detection [24]. Since the size of CSF lies in a certain range, we did not have the problem of detecting troublesome small objects and chose the YOLO v3 for the current study. One disadvantage of the proposed models is that the object detection module (YOLO v3) and the semantic segmentation module (U-net or U-net++) are trained separately. However, the disadvantage of dual modules carries one major strength. In the field of image processing, object detection and semantic segmentation are two critical topics. In this paper, we integrated two tasks mentioned above into one system in the form of two individual modules. Instead of combining those two tasks into one single model, the proposed dual-module architecture provides the flexibility of choosing the proper algorithms and of fine-tuning mechanisms to individually improve the performance of object detection and semantic segmentation.

Physiologically, abdominal compression and hyperventilation could change spinal CSF volume [25]. Spinal CSF volume also influences the effectiveness of spinal anesthesia [26]. Diseases regarding CSF homeostasis, such as normal pressure hydrocephalus (NPH), SIH, idiopathic intracranial hypertension (IIH), or even redistribution of CSF after lumbar CSF withdrawal or intrathecal injection, could be well understood if we had reliable and efficient spinal CSF quantitative methods. While there are many tools for intracranial CSF segmentation, e.g., SPM and FMRIB Software Library (FSL), quantification of spinal CSF volumes from MR imaging is not readily available and is more challenging because of the length of the spinal canal, which necessitates the use of multiple overlapping acquisitions with potentially varying image nonuniformity and lack of a priori tissue probability maps (TPMs), which is the basis of SPM and FSL [27].

After reviewing the literature, no articles using an AI model to segment spinal CSF were found. Considering spinal CSF quantification, Alperin et al. [28] enrolled 10 subjects (2 HVs and 10 IIH patients) and used active contour to segment the 3D T2WI isotropic spinal MR images. They found that, after lumbar CSF withdrawal, a reduction in spinal CSF volume resulted in a drop in intracranial pressure without changes in intracranial CSF in IIH patients. Lebret et al. [29] enrolled 38 subjects (12 HVs and 26 hydrocephalus patients) and used topological assumption with 2660 slices of spinal 3D SPACE MR images to measure spinal CSF. Different ratios of subarachnoid volume to ventricular volume between HVs and hydrocephalus patients were found. In our previous study [10], we used entropy-based segmentation to measure spinal CSF in SIH patients and found that the obtained CSF volume correlated with clinical status. However, the segmentation performance of these articles is not verified with annotated images. In our cohort study, the spinal CSF quantitative results by cascade AI model in SIH patients were correlated with disease status and with the currently used semi-quantitative method, counting the spinal CSF leakage numbers. The obtained spinal CSF volume of our cohort study was comparable to the previously reported whole-spine CSF [28,29,30]. We also observed individual variation in spinal CSF volume in our study. In our previous study, we found that the total CSF amount correlated positively with patient body height. An inverse correlation of lumbosacral CSF volume with BMI has been reported [31,32]. However, further analysis was not performed, due to limited patient numbers.

Several limitations of this study need to be mentioned. First, we only included patients from a single institution, using the same imaging protocol. Our results could be undermined by the MRM protocol. We used 3D SPACE for MRM in our hospital. It is a heavy-T2WI image with high CSF–tissue contrast. However, due to the long TE effect, uneven signals might occur, leading to poor imaging quality. Using the cascade semantic segmentation for other 3D MRM protocols might lead to different performance levels. Our study might provide a basis for future research and software development for general situations. Second, due to the small size of the cohort study, it would not be feasible to analyze the individual differences in spinal CSF volume between patients. Applying our cascade semantic segmentation model to a large number of subjects, different MRM protocols, images from other institutions, and different MR machines are needed in future studies.

## 5. Conclusions

The results show that the cascade AI model provides satisfactory performance in the fully automatic segmentation of spinal CSF from MRM images. The measured spinal CSF volume correlates well with spinal leakage levels and could reflect clinical status in SIH patients. Using this AI model to segment whole-spine CSF, improve diagnosis, evaluate treatment efficacy, and better understand the pathophysiology of diseases regarding CSF homeostasis, such as hydrocephalus, IIH, and SIH, is possible.

## Figures and Tables

**Figure 1 biomedicines-10-02049-f001:**
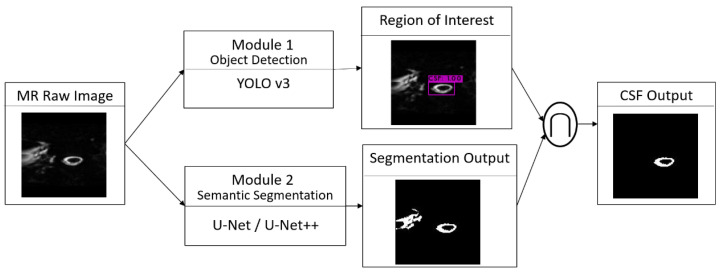
Cascade model. Prediction is combined by using Boolean AND (∩) operation.

**Figure 2 biomedicines-10-02049-f002:**
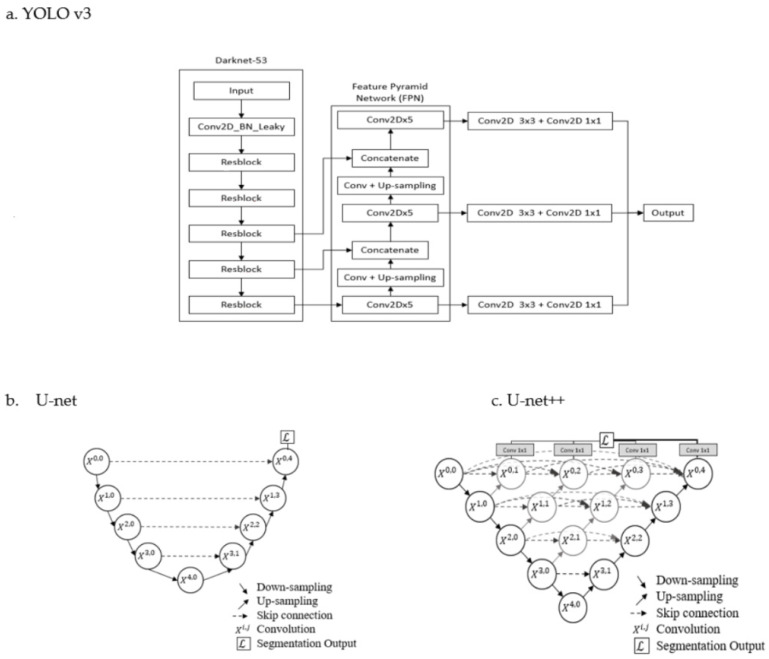
Architectures of YOLO v3 (**a**), U-net (**b**), and U-net++ (**c**) [12,18,19,20].

**Figure 3 biomedicines-10-02049-f003:**
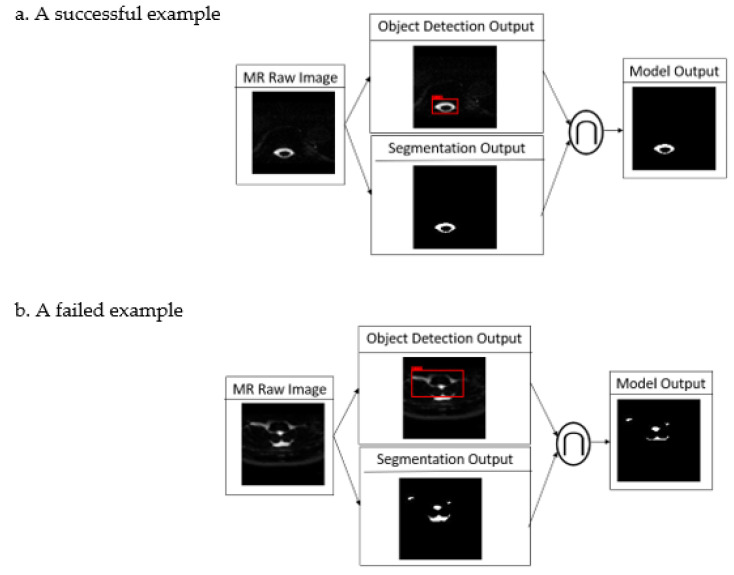
Successful (**a**) and failed (**b**) examples of the proposed cascade model.

**Table 1 biomedicines-10-02049-t001:** Demographics of the participants in the training, validation, test, and cohort study.

Variables	Training and Validation Dataset		Test Dataset		Cohort Study
	Patients	HVs	Patients	HVs	Patients
No.	7	6	6	6	13
Age (mean ± SD)	45.14 ± 9.70	38.00 ± 9.70	41.17 ± 8.75	36.33 ± 6.09	43.57 ± 12.12
Age range (years)	30–61	30–57	30–53	29–46	22–61
Male	2	1	4	1	2
Female	5	5	2	5	11

HVs healthy volunteers; SD standard deviation.

**Table 2 biomedicines-10-02049-t002:** Results of the performance of different AI models.

Algorithms	Mean of IoU	SD of IoU
**Cascade Models—Integration of Object Detection Module and Semantic Segmentation Module**		
1. U-net++ and YOLO v3 (YOLO v3 ∩ U-net++)	0.9374	0.0159
2. U-net and YOLO v3 (YOLO v3 ∩ U-net)	0.9373	0.0158
**Non-cascade Models—Semantic Segmentation Module Only**		
1. U-net++	0.9102	0.0774
2. U-net	0.9077	0.0799

YOLO v3 You Only Look Once version 3, SD standard deviation.

**Table 3 biomedicines-10-02049-t003:** Paired *t*-test of IoU (α = 0.05) between the different cascade models and non-cascade models.

Algorithms	Mean	SD	95% Confidence Int.		DoF	Sig. Level	Results
			Lower Limit	Upper Limit			
**Cascade Models vs. Non-cascade Models**							
YOLO v3 ∩ U-net (IoU_1_) vs. U-net (IoU_2_)	0.0303	0.0834	0.0288	0.0317	12,209	0.000	Accept *H_1_:* IoU_1_ > IoU_2_
YOLO v3 ∩ U-net++ (IoU_1_) vs. U-net++ (IoU_2_)	0.0276	0.0811	0.0262	0.0291	12,209	0.000	Accept *H_1_:* IoU_1_ > IoU_2_
**Comparison of Algorithms in Non-cascade Mod** **els**							
U-net++ (IoU_1_) vs. U-net (IoU_2_)	0.0028	0.0351	0.0021	0.0034	12,209	0.000	Accept *H_1_:* IoU_1_ > IoU_2_

YOLO v3 You Only Look Once version 3, SD standard deviation, IoU metric of intersection over union, Sig. Significance.

## Data Availability

The data that support the findings of this study are available from the corresponding author upon reasonable request.

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
