# Peer review of "Quantitative Measurement of Spinal Cerebrospinal Fluid by Cascade Artificial Intelligence Models in Patients with Spontaneous Intracranial Hypotension"

_biomedicines, 2022, doi:10.3390/biomedicines10082049_

Round 1

Reviewer 1 Report

In this study the authors aimed to elaborate a cascade artificial intelligence (AI) module to automatically quantify the spinal Cerebrospinal fluid (CSF). They tested their method in 13 patients with spontaneous intracranial hypotension (SIH) and in 12 healthy controls. AI module provided good performance and the obtained spinal CSF volume could reflect the clinical status. 

Statistical analyses have well conducted and figures and tables are informative.

The paper is interesting but I have some concerns:

  1. Table 1 is cut off and the last column is not readable.
  2. The simple size is very small and the all the subjects come from the same institution with the same sequence. This is very unusual for an AI module validation study.
  3. The different groups are not well matched by age and gender.
  4. Why the authors do not mention automatic segmentation software that calculates CSF volumes such as SPM or FSL's SIENAX tool? A comparison would also have been interesting.

Reviewer 2 Report

The problem discussed in the paper is very interesting. The paper is well-written. However, the paper has the following concerns:

Contributions need to be highlighted in detail as authors have combined two well existing methods. 

Recent related papers need to be studied and included.

Define IoU in the abstract.

Why did authors choose YOLO v3 and UNet? Why not other popular CNN based segmentation models? Comment on it.

Discuss about the architecture of both the networks and the hyperparameter settings.

State the weaknesses and strengths of your model.

Can you show one scenario where your model fails? You can provide few subjective results of your model here.

Conclusion section seems very short.

Paper has few language and grammar errors.  

Round 2

Reviewer 1 Report

The paper in present form is ready for publication.